# Methods and Characteristics of Drug Extraction from Ion-Exchange-Resin-Mediated Preparations: Influences, Thermodynamics, and Kinetics

**DOI:** 10.3390/polym15051191

**Published:** 2023-02-27

**Authors:** Junlin Yuan, Conghui Li, Shanshan Wang, Hui Zhang, Zengming Wang, Aiping Zheng, Xiuli Gao

**Affiliations:** 1School of Pharmacy, Guizhou Medical University, Guiyang 550025, China; 2State Key Laboratory of Toxicology and Medical Countermeasures, Beijing Institute of Pharmacology and Toxicology, 27th Taiping Road, Beijing 100850, China

**Keywords:** ion-exchange resin, counterions, thermodynamics, kinetics, methylphenidate hydrochloride

## Abstract

Since the discovery of ion-exchange resins, they have been used in many fields, including pharmacy. Ion-exchange resin-mediated preparations can realize a series of functions, such as taste masking and regulating release. However, it is very difficult to extract the drug completely from the drug–resin complex because of the specific combination of the drug and resin. In this study, methylphenidate hydrochloride extended-release chewable tablets compounded by methylphenidate hydrochloride and ion-exchange resin were selected for a drug extraction study. The efficiency of drug extraction by dissociating with the addition of counterions was found to be higher than other physical extraction methods. Then, the factors affecting the dissociation process were studied to completely extract the drug from the methylphenidate hydrochloride extended-release chewable tablets. Furthermore, the thermodynamic and kinetic study of the dissociation process showed that the dissociation process obeys the second-order kinetic process, and it is nonspontaneous, entropy-decreasing, and endothermic. Meanwhile, the reaction rate was confirmed by the Boyd model, and the film diffusion and matrix diffusion were both shown to be rate-limiting steps. In conclusion, this study aims to provide technological and theoretical support for establishing a quality assessment and control system of ion-exchange resin-mediated preparations, promoting the applications of ion-exchange resins in the field of drug preparation.

## 1. Introduction

Ion-exchange resins are a class of porous polymers composed of three parts: counterions, functional groups, and three-dimensional structures [1]. Counterions bound to functional groups by ionic bonds play a major role in the ion-exchange process [2,3,4,5]. Ion-exchange resins exhibit various advantages by exchanging with ionic drugs and allowing the drugs to enter the ion-exchange resin backbone to form drug–resin complexes [6,7,8,9]. As a result, they are utilized in many roles in the field of pharmacy, for example, in controlled release, solubilized, and taste-making systems.

Methylphenidate hydrochloride (MPH) is a derivative of piperidine acetate, which has the effects of stimulating the central nervous system, reducing fatigue, relieving depression, etc. However, the bitter taste, short half-life, and low bioavailability of MPH reduce its clinical efficacy [10,11]. Extended-release chewable tablets prepared with ion-exchange resin can maintain the extended-release property during chewing, which can simultaneously achieve drug-taste masking and reduce the frequency of drug administration, thereby improving compliance and therapeutic effects.

In methylphenidate hydrochloride extended-release chewable tablets, the methylphenidate cation forms a drug–resin complex by replacing counterions (Na^+^) in the ion-exchange resin (Amberlite^®^ IRP 69) to mask the taste and regulate release (Figure 1) [12]. Three interactions between MPH and Amberlite^®^ IRP 69 are known from molecular dynamics simulations (Figure 2): the hydrogen bonds formed between the nitrogen atoms in the six-membered cyclic hydrocarbon of MPH and the sulfonic acid group of the resin; the CH3–π interactions formed between the methyl group of MPH and the benzene ring of the resin; the π–π interactions formed between the benzene ring of MPH and the benzene ring of the resin; and, most importantly, the salt bridges between the nitrogen atoms in the six-membered cyclic hydrocarbon of MPH and the sulfonic acid group of the resin [12]. Since the drug cations and the resin are mainly bound by salt bridges, it is not feasible to extract the drug by conventional physical methods.

This is one of the difficulties in completely extracting a drug from ion-exchange resin-mediated formulations due to the lack of systematic research on the mechanism of ion-exchange resin-mediated formulations. So far, there is no effective method of completely extracting drugs from ion-exchange resin-mediated formulations. However, since ion-exchange resins play an important role in drug formulation and delivery systems, it is important to completely extract drugs from them. Additionally, the inability to completely extract the drug makes it difficult to establish a safe and effective quality control system for ion-exchange resin-mediated formulations. Therefore, in this study, MPH was used as a model drug to investigate, in depth, how to completely extract the drug from the ion-exchange resin-mediated formulation. From previous studies, we know that the compounding process of MPH and Amberlite^®^ IRP 69 is a reversible ion-exchange process. MPH cations are combined on the resin to form a drug–resin complex by exchanging sodium ions [12]. Conversely, the MPH cation combined on the resin can also be dissociated by the competition of other cations in the system. Based on the above principles, a method for the complete extraction of drugs from methylphenidate hydrochloride extended-release chewable tablets can be established.

In this study, a method for the complete extraction of drugs from ion-exchange resin-mediated formulations was innovatively established based on the ion-exchange principle, using methylphenidate hydrochloride extended-release chewable tablets as a model and the recovery rate as an index. Based on this drug extraction, an accurate and sensitive assay method was established to achieve good quality control. Furthermore, the factors affecting the dissociation process were studied, and the reaction mechanism of the process was discussed from the thermodynamic and kinetic perspectives. In conclusion, the purpose of this study is to provide technological and theoretical support for the establishment of a quality assessment and control system of ion-exchange resin-mediated preparations, which will promote the application and extension of ion-exchange resins in the field of drug preparation.

## 2. Materials and Methods

### 2.1. Materials

The model drug methylphenidate hydrochloride extended-release chewable tablets were donated from the Institute of Pharmacology and Toxicology, Beijing, China. Amberlite^®^ IRP69, namely sodium polystyrene sulfonate (complex polymer), which is a strongly acidic cation-exchange resin, was purchased from DuPont (Wilmington, DE, USA). The reagents covered in this article, such as diluted hydrochloric acid (HCl), sodium chloride (NaCl), potassium chloride (KCl), and phosphoric acid (H_3_PO_4_), are all analytical-grade reagents (Sinopharm Chemical Reagent Co., Ltd., Shanghai, China).

### 2.2. Methods

#### 2.2.1. Chromatographic Conditions

The separation was performed at 30 °C using an analytical column of C18 (C18, 3.9 × 150 mm, 5 μm, Waters, Milford, MA, USA). The mobile phase, at the flow rate of 1 mL/min, consisting of an acetonitrile/phosphate sodium octane sulfonate solution (12 mM, pH = 2.7) with a ratio of (27/73 *v*/*v*) was applied in the isocratic mode. The examination wavelength was 210 nm and the injection volume used was 25 μL.

#### 2.2.2. Study of Extraction Method

The appropriate number of methylphenidate hydrochloride extended-release chewable tablets were dissolved in a 50 mL volumetric flask and the appropriate amount of diluent was added for the total volume. Then, the drug was extracted by four methods: ultrasonic extraction, high-pressure homogenization, ultrasonic cell crusher, and dissociation by counterions. Finally, the mixture was diluted with diluent to the appropriate volume. Recovery was used as an index to assess the methods of drug extraction, and the details of the extraction methods are shown in Table 1.

#### 2.2.3. Study of Factors Influencing the Dissociation Process

To completely extract the drug, we studied the factors affecting the dissociation process by varying four independent variables: the type of counterions, the concentration of counterions, the ratio of acetonitrile, and the dissociation temperature, and the parameters of each variable are shown in Table 2 and Figure 3. Data are expressed as the mean ± SD. Meanwhile, *t*-tests were used for data analysis. The results of this study will provide guidance on the best factors to use to extract the drug from methylphenidate hydrochloride extended-release chewable tablets.

#### 2.2.4. Analytical Method Validation

The assay method of methylphenidate hydrochloride extended-release chewable tablets was validated according to ICH Q2 (R1) guidelines. Five items were validated: specificity, linearity and range, accuracy, repeatability, and intermediate precision. The details are listed in Table 3.

#### 2.2.5. Study of the Thermodynamics of Dissociation

The equation for the dissociation process of MPH from ion-exchange resin is as follows:(1)$$R-SO3-MPH^{+}+D^{+}↔R-SO3-D^{+}+MPH^{+}\;$$
where *R-SO3-MPH^+^* is the drug–resin complex and *D^+^* is the counterion. When the ion-exchange reaction reaches equilibrium, the reaction equilibrium relationship can be described according to the law of mass action as [13]:(2)$$K_{e}=\frac{\left[D^{+}]\;r[MPH^{+}\right]s}{\left[D^{+}]\;s[MPH^{+}\right]r}$$
where [*D^+^*]_r_ and [*MPH*^+^]_r_ (mmol/g) are the equilibrium concentrations of the *D^+^* and *MPH*^+^ in the resin, separately, and [*D*^+^]_s_ and [*MPH*^+^]_s_ (mmol/L) are the equilibrium concentrations of the *D*^+^ and *MPH*^+^ in solution. The equilibrium constant, *Ke*, can be used to describe the degree of chemical reaction. The standard free energy change Δ*G* can be calculated by the following equation:(3)ΔG=−RTlnK

The enthalpy change Δ*H* can be calculated based on the Van’t Hoff equation:(4)$$\ln k=-\frac{\Delta H}{RT}-C$$

The entropy change Δ*S* can be calculated based on the Gibbs–Helmholtz equation:(5)$$\Delta S=-\frac{\Delta H-\Delta G}{T}$$

#### 2.2.6. Study of the Kinetics of Dissociation

##### Simple-Order Reactions

Based on the basic theory of chemical kinetics, the simple level reaction equation for the dissociation of MPH from methylphenidate hydrochloride extended-release chewable tablets is expressed as follows [9]:

Zero-order reaction:(6)$$Q_{e}-Q_{t}=-kt$$

First-order reaction:(7)$$\ln\left(Q_{e}-Q_{t}\right)=-kt$$

Second-order reaction:(8)$$\frac{1}{Q_{e}-Q_{t}}=kt$$
where $Q_{e}$ and $Q_{t}$ are the recoveries of MPH at equilibrium at *t*, *k* is the reaction rate constant, and *t* is the reaction time (min).

##### Model of Kinetics

Ion-exchange processes are described by three kinetic models: the shrinking core model (SCM), the homogeneous diffusion model (HDM), and the Boyd model (Boyd) [14,15,16,17,18].

Shrinking core model (SCM)

This approach is suitable when the porosity of the resin is small. If film diffusion is the controlling step, then the following equation is valid:(9)$$F=\frac{3C_{Ao}K_{mA}}{ar_{0}C_{so}}t$$
where $r_{0}$ is the diameter of the resin, $C_{Ao}$ is the molar concentration of counterions in the solution, $C_{so}$ is the molar concentration of a solid reactant at the bead’s interacted core, a is the stoichiometric coefficient, and $K_{mA}$ is the mass transfer coefficient of species A passing through the liquid membrane.

When matrix diffusion is the controlling step, then the following equation is valid:(10)$$3-3{\left(1-F\right)}^{\frac{2}{3}}-2F=\frac{6D_{er}C_{Ao}}{ar_{0}^{2}C_{so}}t$$
where $D_{er}$ is the diffusion coefficient in the solid phase in (m^2^/s).

When the chemical reaction is the controlling step, the following equation is applicable:(11)$$1-{\left(1-F\right)}^{\frac{1}{3}}=\frac{k_{s}C_{Ao}}{r_{0}}t$$
where $k_{s}$ is the reaction constant, which is based on the surface in (m/s).

2.Homogeneous diffusion model (HDM):

The HDM model assumes that the rate of the chemical reaction is much greater than the rate of diffusion. Meanwhile, the resin particle is regarded as a regular sphere. If the matrix diffusion is the controlling step, the following equation is valid:(12)$$-\ln\left(1-F^{2}\right)=2\pi^{2}\frac{D_{r}}{r_{0}^{2}}t$$
where F is the extent of conversion, $D_{r}$ is the diffusion coefficient in the solid phase (m^2^/s), $r_{0}$ is the particle radius in (m), and t is the time in (s).

The following expression can be applied if liquid film diffusion is the controlling step: (13)$$-\ln\left(1-F\right)=\frac{3D}{\delta r_{0}}\frac{C}{C_{r}}t$$
where D is the diffusion coefficient in the solution phase (m^2^/s), C is the total molar concentration of the exchanging species, $C_{r}$ is the molar concentration of the exchanging species in the ion-exchange phase, and δ is the thickness of the liquid film in (m).

3.Boyd model (Boyd)

Ion-exchange kinetics are often described by the Boyd model. The model is as follows:(14)$$F=1-\frac{6}{\pi^{2}}\overset{\infty }{\underset{n=1}{\sum }}\frac{e^{-n^{2}Bt}}{n^{2}}$$
where F is the percentage of adsorption at time *t*, B (min^−1^) is the constant of the exchange rate, and n is the sum of variables. If the value of *F* exceeds 0.85, the equation can be simplified to the following shape:(15)$$Bt=-2.303\log_{10}\left(1-F\right)-0.498$$

When the value of F is lower than 0.85, the equation can be simplified as follows:(16)$$Bt=6.283-3.290F-6.283{\left(1-1.047F\right)}^{1/2}$$

## 3. Results

### 3.1. Study of Extraction Method

The recoveries measured for the samples treated with different extraction methods are shown in Figure 4, and the statistical analysis is displayed in Table 4. From the results, it is clear that *p*-values from two-sample *t*-tests are less than 0.05, which indicates there are significant differences between the recoveries of physical extraction methods (ultrasonic extraction, high-pressure homogenization, and ultrasonic cell crusher) and the recoveries of dissociation. It was obvious that the drug could not be completely extracted using ultrasonic extraction, high-pressure homogenization, or the ultrasonic cell crusher, and the recoveries were all lower than 20%. However, when the 1.0 mol/L HCl solution was used for the dissociation treatment, the recovery increased significantly, reaching more than 80%. This result is consistent with the ion-exchange principle mentioned in the previous section. Therefore, dissociation was chosen as the drug extraction method and the factors influencing the dissociation process were further investigated.

### 3.2. Study of Factors Influencing the Dissociation Process

#### 3.2.1. Type of Counterions

The results of the recoveries in different species of counterion systems for the same quantities of counterions are shown in Figure 5A. The highest recovery was obtained when the counterion was K^+^. This indicated that the dissociation effect of K^+^ was better than Na^+^ and H^+^ when the number of counterions was the same. Therefore, KCl was chosen as the dissociation media for further study.

#### 3.2.2. Concentration of Counterions

As shown in Figure 5B and Table 5, the *p*-values indicate that differences between the recoveries of 2.5 mol/L and the recoveries of other concentrations are statistically significant. This demonstrates that when the counterions in the solution were the same and both were potassium ions, recoveries increased with an increase in counterion concentration. This indicates that when the MPH concentration was certain, the quantities of counterions in the system increased as the concentration of it increased; therefore, the quantity of MPH being dissociated increased and the drug extraction was more complete. Thus, the counterions concentration of 2.5 mol/L was chosen.

#### 3.2.3. Ratio of Acetonitrile

As shown in Figure 5C and Table 5, the *p*-values indicate that differences between the recoveries of 0% acetonitrile and the recoveries of 20% acetonitrile are statistically significant. This suggests that increasing the proportion of acetonitrile can improve the recoveries. When only an inorganic salt solution was used as the dissociation media, it was difficult to dissociate the drug completely, and the recovery was always lower than 98%. However, when a certain amount of acetonitrile was added to the dissociation media, the recovery was significantly improved, and the recovery could reach the range of 98–102%. Therefore, it is necessary to add a certain amount of acetonitrile to the dissociation media to facilitate the extraction of drugs.

#### 3.2.4. Dissociation Temperature

Based on the results at different temperatures, as shown in Figure 5D and Table 5, it is clear that the *p*-values from the two-sample *t*-tests of the recoveries at 30 °C and 50 °C were less than 0.05, which indicates there are significant differences between the recoveries at 30 °C and 50 °C. However, the *p*-values of the two-sample *t*-tests from the recoveries at 40 °C and 50 °C are greater than 0.05, indicating that there are no significant differences between the recoveries at 40 °C and 50 °C. The above results indicate that when the temperature increased, the recovery increased correspondingly, indicating that increasing the temperature helped the dissociation of the drug. However, when the temperature reached 40 °C, the recovery reached 98–102% and the drug was extracted completely, so the recoveries remained stable and did not increase any further. Considering that the drug can be completely extracted at 40 °C and the energy consumption is lower than at 50 °C, a dissociation temperature of 40 °C was chosen.

Finally, the extraction method was determined as dissociation. Meanwhile, the type of counterions was determined as K^+^, the concentration of counterions was determined as 2.5 mol/L, the ratio of acetonitrile was determined as 20%, and the dissociation temperature was determined as 40 °C. MPH can be completely extracted from methylphenidate hydrochloride extended-release chewable tablets by applying the above process, and the recovery rate can reach the range of 98–102%.

### 3.3. Analytical Method Validation

The validation of the analytical method was performed to demonstrate that it was suitable for the assay of methylphenidate hydrochloride extended-release chewable tablets. The selected analytical method was validated for each parameter in terms of specificity, linearity and range, accuracy, repeatability, and intermediate precision. The results are shown in Table 6 and Figure 6. The results showed that the blank solvent and blank excipients did not interfere with the determination of MPH, and the separation of MPH and neighboring peaks in the degradation products was >1.5; the relationship between peak area and concentration of MPH in the range of 0.06 mg/mL–0.32 mg/mL was Y = 0.6461X + 0.7529 with a correlation coefficient of 0.9999, which was >0.999; the average value of the six MPH solutions was 94.34% with an RSD of 0.17%, which was less than 2.0%; the recoveries of MPH at all concentrations were in the range of 95.0~105.0%, and the RSDs were less than 2.0%. The above results show that the results of all analytical method validation items comply with the standards of ICH Q2 (R1) guidelines. This indicates that the method is suitable for the assay of MPH from methylphenidate hydrochloride extended-release chewable tablets.

### 3.4. Study of the Thermodynamics of Dissociation

Thermodynamics mainly describes the energy change, direction, and degree of a reaction. A larger equilibrium constant *K_e_* indicates that the degree of ion-exchange reaction is closer to completion. At different temperatures, the calculated values of *K*_e_ and other thermodynamics parameters are shown in Table 7. With increasing temperature, *K*_e_ increased. This indicated that a high temperature was favorable for the reaction proceeding in the forward direction. The enthalpy change Δ*H* > 0 indicates that the reaction was endothermic. Therefore, a higher temperature helps the reaction to proceed completely. The standard free energy change Δ*G* > 0 proved that the reaction was nonspontaneous. Moreover, the entropy change Δ*S* < 0 suggested that the movement of the MPH cation was limited after dissociating from the methylphenidate hydrochloride extended-release chewable tablets [19].

### 3.5. Study of the Kinetics of Dissociation

#### 3.5.1. Simple-Order Reactions

At 298 K, the reaction levels of the ion-exchange reaction were determined based on the fitting of the recovery-time curves using the zero-order, first-order, and second-order kinetic equations. The fitting results, shown in Table 8, indicate that the dissociation of MPH from methylphenidate hydrochloride extended-release chewable tablets was consistent with second-order kinetics. *Q_e_* and *Q_t_* denote the recovery at equilibrium and time *t*, and *k* is the constant of the reaction rate. According to the principle of chemical reaction, the dissociation reaction is a second-order reversible reaction [13].

#### 3.5.2. Model of Kinetics

As shown in Figure 7, the dissociation of MPH from methylphenidate hydrochloride extended-release chewable tablets can be explained by a series of rate-limiting steps: (a) ion diffusion through the liquid film which surrounds the resin; (b) the diffusion of ions through the matrix within the resin particles; and (c) the chemical reaction with functional groups attached to the matrix. If one of the steps generally provides more resistance than the others, it can be considered the rate-limiting step of the dissociation process [20,21,22]. The models widely used to describe the ion-exchange kinetic process are the shrinking core model (SCM), homogeneous diffusion model (HDM), and Boyd model (Boyd). Table 9 and Figure 8 conclude the linear regression analysis using SCM, HDM, and the Boyd model for the dissociation process. Good correlation coefficients (r) were obtained from HDM and Boyd, indicating that the HDM model and Boyd model were more suitable for describing the ion-exchange reaction of MPH dissociation from methylphenidate hydrochloride extended-release chewable tablets. Furthermore, the fitting results of the two rate-limiting steps in the HDM model and the Boyd model were very similar, and the correlation coefficients could reach above 0.98 for both. This indicated that both film diffusion and matrix diffusion are rate-limiting steps.

### 3.6. Study of Factors Influencing the Dissociation Process Using the Boyd Model

Based on the study of the ion-exchange kinetic model, the Boyd model was chosen to analyze the different factors affecting the dissociation process. It was studied by plotting *B*t − t, and the magnitude of the reaction rate was obtained from the slope of the line. The kinetics of the factors influencing the dissociation process of MPH from methylphenidate hydrochloride extended-release chewable tablets can be carried out by the magnitude of the reaction rate.

#### 3.6.1. Type of Counterions

The fitting results of the types of counterions are listed in Table 10 and Figure 9. When the concentrations of counterions were consistent, the reaction rates were similar when the counterions were H^+^ and K^+^. However, the rate of the dissociation of MPH from methylphenidate hydrochloride extended-release chewable tablets was significantly reduced when the counterion was Na^+^.

#### 3.6.2. Concentration of Counterions

The fitting results of the types of counterions are shown in Table 11 and Figure 10. When counterion types were the same and both were potassium ions, the reaction rate decreased as the quantities of counterions increased. From the previous section, it is clear that as the concentration of counterions increased, the quantities of dissociated MPH increased, which facilitated drug extraction, but decreased the rate of MPH dissociation from methylphenidate hydrochloride extended-release chewable tablets.

#### 3.6.3. Ratio of Acetonitrile

As shown in Table 12 and Figure 11, when a certain amount of acetonitrile was added to the dissociation medium, the reaction rate increased. Combined with the previous results, the addition of acetonitrile not only improved the degree of MPH dissociation from methylphenidate hydrochloride extended-release chewable tablets but also increased the rate of the dissociation reaction.

#### 3.6.4. Dissociation Temperature

As shown in Table 13 and Figure 12, reaction rates at different temperatures were similar in the dissociation process. This indicates that for the reaction of dissociation of MPH from methylphenidate hydrochloride extended-release chewable tablets, the temperature had less effect on the reaction.

## 4. Discussion

The ultrasonic extraction, high-pressure homogenization, and ultrasonic cell crusher methods were not able to extract MPH from methylphenidate hydrochloride extended-release chewable tablets effectively because all of the above methods are forms of physical extraction. However, MPH and Amberlite^®^ IRP69 are not combined by physical interactions alone in a drug–resin complex. There are three interactions between MPH and Amberlite^®^ IRP69 [12]. Firstly, there are hydrogen bonds between MPH and Amberlite^®^ IRP69, mainly formed by the interaction of the nitrogen atoms in the six-membered ring hydrocarbon of the MPH with the sulfonic acid groups of the resin. Secondly, there are two types of stacking interactions between MPH and Amberlite^®^ IRP69, which are the CH3–π interaction formed by the methyl group of MPH and the benzene ring of the resin, and the π–π interaction formed by the benzene ring of MPH and the benzene ring of the resin. The last and most important interaction is the salt bridge between the nitrogen atom in the six-membered ring hydrocarbon of MPH and the sulfonic acid group of the resin [12].

The effect of drug extraction was increased by adding counterions to the system for dissociation. This is because the counterions competed with MPH for the salt bridge interaction between MPH and Amberlite^®^ IRP69, decreasing their interaction energy and thus weakening the combination between MPH and Amberlite^®^ IRP69, which makes it easy to dissociate MPH from the drug–resin complex [12]. Water acts as a hydrogen bonding donor in the bonding process between MPH and resin, and the ability of water as a protic solvent to provide hydrogen bonds is greater than that of acetonitrile [23,24]. When a certain amount of acetonitrile is added, the amount of water is relatively reduced and the ability to provide hydrogen bonding is weakened. Therefore, the hydrogen bond between MPH and Amberlite^®^ IRP69 was broken, thus weakening the link between MPH and resin, so the dissociation effect was further improved.

When the concentration of counterions was the same, the dissociation effect increased in the order of H^+^, Na^+^, and K^+^. According to the study of the complexation process of MPH and Amberlite^®^ IRP69, the interaction energy between the counterions and the functional groups attached to Amberlite^®^ IRP69 increased in the order of H^+^, Na^+^, and K^+^ [12]. K^+^ can combine with the functional groups more efficiently, thus effectively inhibiting the association between MPH and Amberlite^®^ IRP69, which makes it easier for MPH to dissociate. Meanwhile, the dissociation effect of MPH increased with the increase in counterion quantities when the counterion types were the same. This is because as the quantities of counterions increased, the number of combinations between counterions and functional groups also increased, thus inhibiting the association between MPH and Amberlite^®^ IRP69 more effectively.

Thermodynamic considerations of the ion-exchange process are necessary to determine the extent and spontaneity of the process, etc., [13,23]. *K_e_* increases with increasing temperature, indicating that increasing temperature facilitates the dissociation reaction. Δ*H* > 0 indicates that the reaction is endothermic and that the elevated temperature contributes to the dissociation reaction. Δ*G* > 0 proves that the reaction is nonspontaneous. In addition, the entropy change Δ*S* < 0 indicates that the system is less chaotic and that the movement of MPH cations is limited after dissociating from methylphenidate hydrochloride extended-release chewable tablets [13]. Increasing the temperature facilitated the dissociation of MPH. This is because, on the one hand, increasing the temperature increased the rate of counterions entering the resin core. On the other hand, the dissociation reaction is endothermic due to the enthalpy change Δ*H* > 0, which leads to an increase in *Ke* with increasing temperature [13]. As a result, the overall extent of the dissociation reaction increased, which led to an increase in the degree of MPH dissociation.

Ion-exchange kinetics mainly includes the study of ion-exchange mechanisms, rate-limiting steps, and factors affecting reaction rate [13,24]. The correlation coefficient was calculated by fitting the simple-order reaction, and the correlation coefficient for the second-order reaction was the largest and greater than 0.99. This indicates that the dissociation reaction is a second-order reaction [23,24]. According to the results of the kinetic model fitting, the correlation coefficient is low when the SCM model was used for fitting, indicating that the SCM model does not apply to the dissociation process of MPH. When the HDM and Boyd models were used for fitting, the correlation coefficient could reach above 0.98, which indicated that the HDM and Boyd models are more suitable for describing the dissociation reaction of MPH [14,15]. By further analyzing the fitting results, it was found that the fitting results of film diffusion and matrix diffusion in the HDM model and matrix in the Boyd model were very similar, and the correlation coefficients could reach above 0.98. This indicated that both film diffusion and matrix diffusion are the rate-limiting steps of the dissociation reaction of MPH. According to the Boyd model fitting results, since the Boyd model suggests that the dissolution structure inside the resin and the liquid film structure on the resin surface are basically the same, this indicated that the diffusion state of MPH cations inside the resin and the diffusion state inside the liquid film is basically the same.

The dissociation reaction rate was studied by plotting *Bt-t* through the Boyd model, with the slope of the line representing the reaction rate [23]. Therefore, the Boyd model was chosen to fit the factors influencing the dissociation process of MPH. The results of the fitting showed that the reaction rate decreased with the increase in counterion concentration. This is because when the amount of resin is constant, the quantities of combining sites between the counterions and the resin, which is the sulfonic acid group, is constant, and as the quantities of counterions increase, the competition between the counterions combining with the resin increases, leading to a decrease in the dissociation reaction rate of MPH [13]. Additionally, with the addition of a certain volume of acetonitrile, the dissociation of MPH was accelerated because the hydrogen bond between MPH and the resin was broken, weakening the binding between the drug and the resin [24,25].

## 5. Conclusions

In this study, different methods were used to extract MPH from methylphenidate hydrochloride extended-release chewable tablets. The results showed that due to the specific combining mode of the drug and the resin, dissociation was superior to the other three physical extraction methods. Then, in the study of the effect on the dissociation process, it was found that selecting K^+^ as the counterion, increasing the concentration of the counterions, increasing the temperature, and adding a certain volume of acetonitrile could make the dissociation process more complete. Finally, the thermodynamics and kinetics of the dissociation processes were studied, which were shown to obey the second-order kinetic process and be nonspontaneous, entropy-decreasing, and endothermic. Meanwhile, the reaction rate was confirmed by the Boyd model, and the film diffusion and matrix diffusion were both rate-limiting steps. Analysis of the factors affecting the dissociation process by the Boyd model revealed that the dissociation reaction rate decreased when the counterions were Na^+^ and when the concentration of the counterions increased, and the dissociation reaction rate increased with the addition of a certain volume of acetonitrile. In conclusion, this study can provide a technical reference and theoretical support for drug extraction from specific preparations mediated by ion-exchange resins for quality assessment and control. It is hoped that this promotes the application of ion-exchange resins in the field of drug preparation.

## Figures and Tables

**Figure 1 polymers-15-01191-f001:**
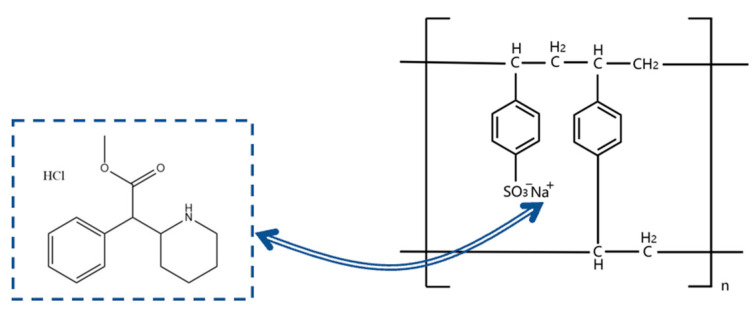
Schematic diagram of the ion-exchange reaction between methylphenidate hydrochloride (MPH) and Amberlite^®^ IRP69.

**Figure 2 polymers-15-01191-f002:**
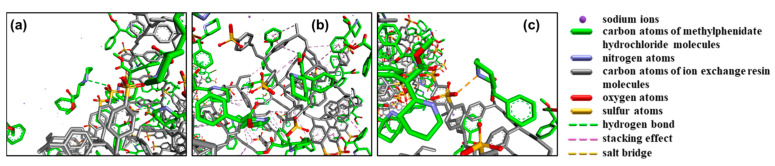
Interaction between MPH and the ion-exchange resin. (**a**) Hydrogen bonding interaction, (**b**) CH3-π and π-π stacking interaction, (**c**) salt bridge interaction.

**Figure 3 polymers-15-01191-f003:**
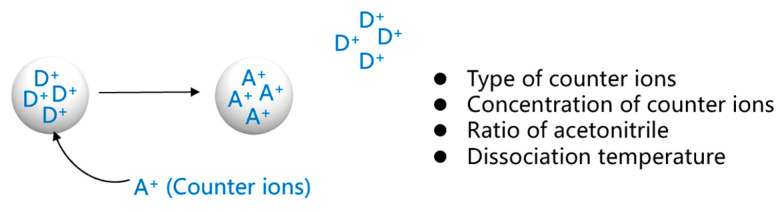
Study of factors affecting the dissociation process. (Where D^+^ is the drug ions on the drug-resin complex and A^+^ is the counter ions).

**Figure 4 polymers-15-01191-f004:**
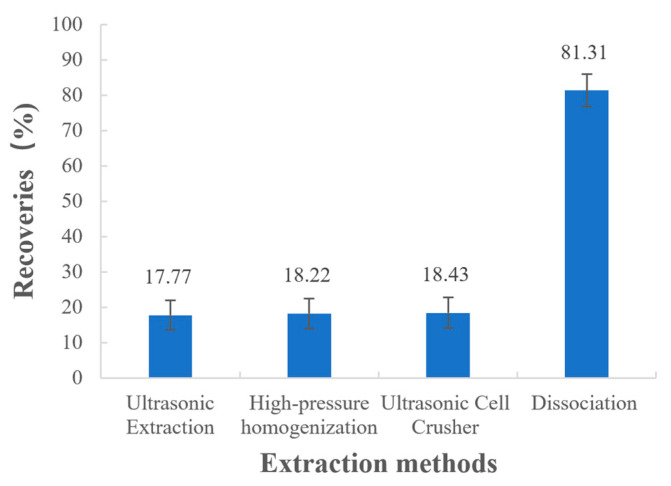
Recoveries of different extraction methods.

**Figure 5 polymers-15-01191-f005:**
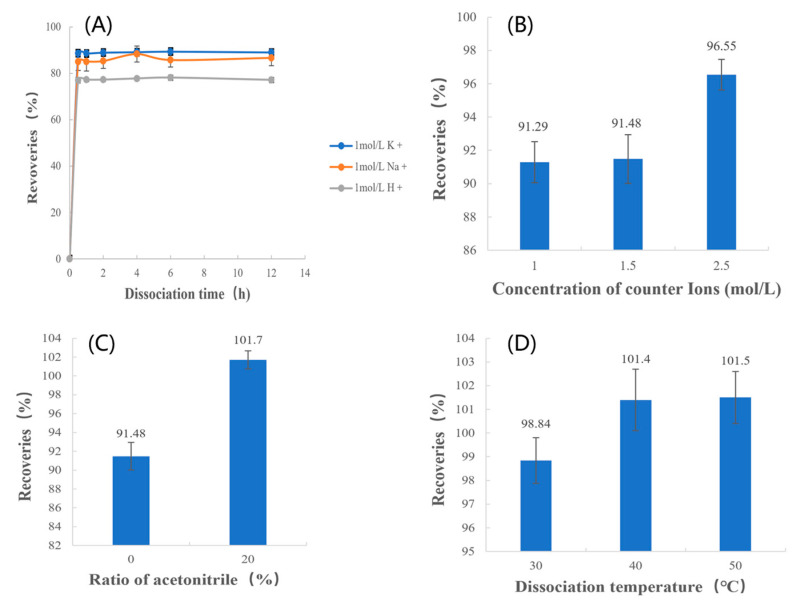
Recoveries of (**A**) type of counterions, (**B**) concentration of counterions, (**C**) ratio of acetonitrile, and (**D**) dissociation temperature.

**Figure 6 polymers-15-01191-f006:**
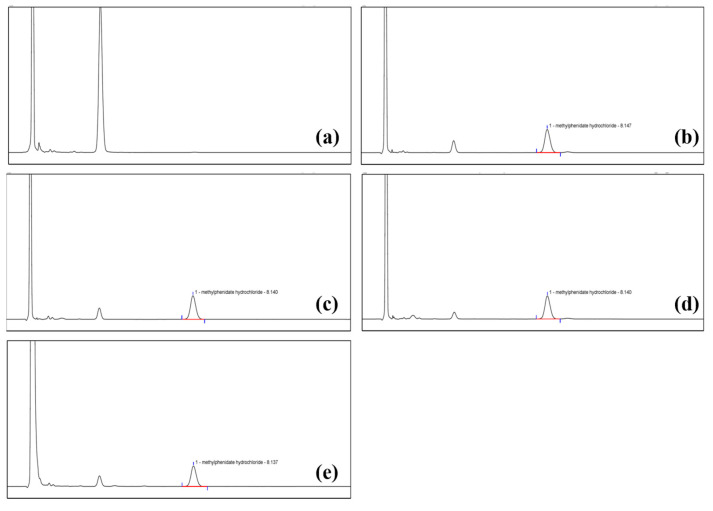
The results of specificity for (**a**) blank excipients, (**b**) test solution, (**c**) degradation products of acid, (**d**) degradation products of the base, and (**e**) degradation products of oxygen.

**Figure 7 polymers-15-01191-f007:**
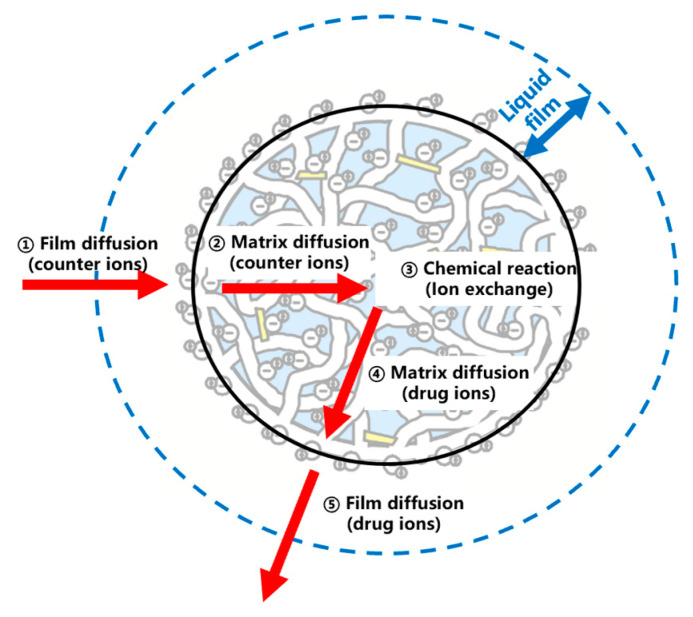
The dissociation process of MPH. (Arrow 1 indicates that the counter ions cross the liquid film, arrow 2 indicates the diffusion of counter ions inside the resin, 3 indicates ion exchange reaction, arrow 4 indicates the diffusion of drug ions inside the resin, arrow 5 indicates that the drug ions cross the liquid film).

**Figure 8 polymers-15-01191-f008:**
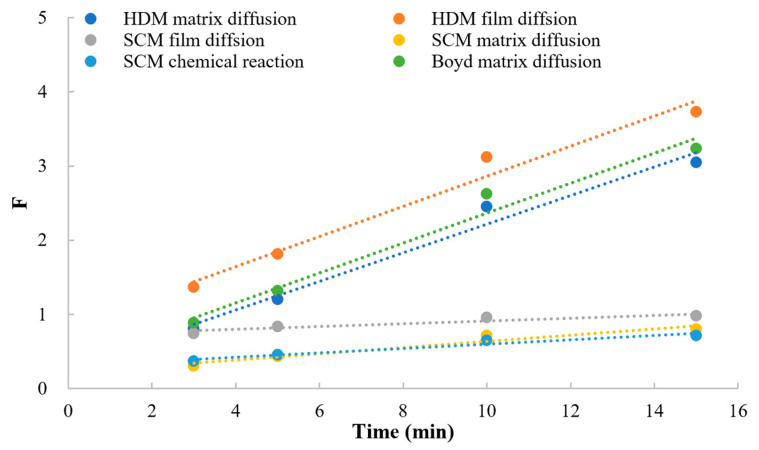
The linear regression analysis for HDM, SCM, and Boyd.

**Figure 9 polymers-15-01191-f009:**
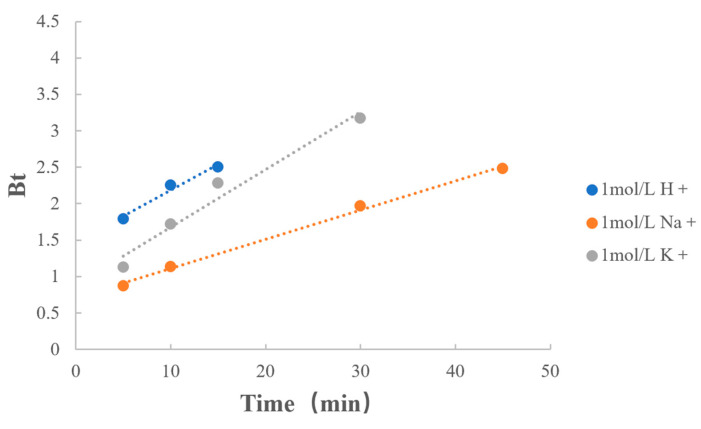
Boyd model fitting on type of counterions.

**Figure 10 polymers-15-01191-f010:**
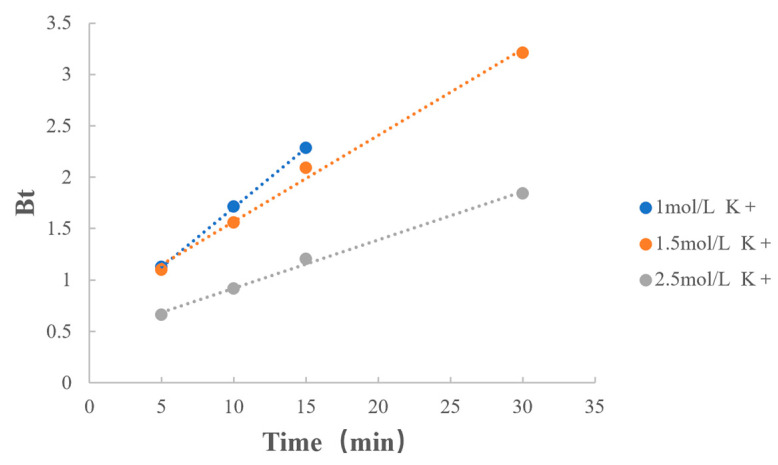
Boyd model fitting on the concentration of counterions.

**Figure 11 polymers-15-01191-f011:**
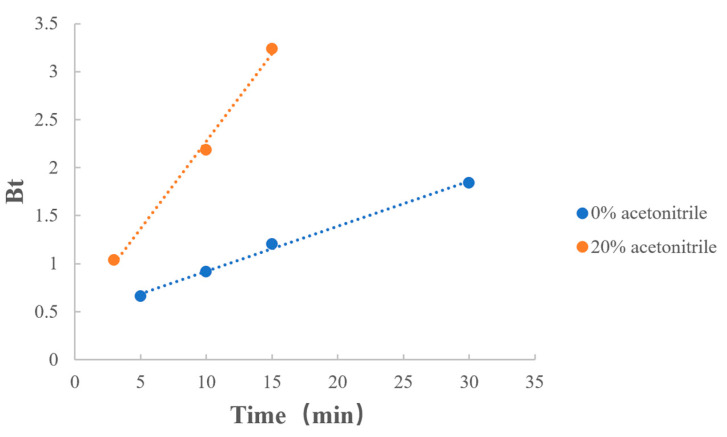
Boyd model fitting on the ratio of acetonitrile.

**Figure 12 polymers-15-01191-f012:**
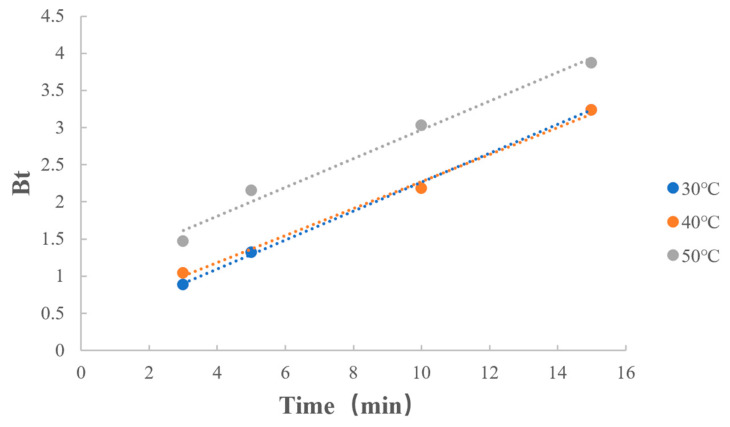
Boyd model fitting on dissociation temperature.

**Table 1 polymers-15-01191-t001:** Study of extraction methods.

Extraction Method	Diluent Type	Sample Extraction
Ultrasonic Extraction	Acetonitrile—pH 3.0 acidified water (25:75)	Ultrasound for 10 min
High-Pressure Homogenization	Acetonitrile–methanol (50:50)	Homogenize for 5 min
Ultrasonic Cell Crusher	Acetonitrile–methanol (50:50)	Process with ultrasonic cell crusher for 5 min
Dissociation by Counterions	1 mol/L hydrochloric acid	Stir for 4 h

**Table 2 polymers-15-01191-t002:** Study of factors affecting the dissociation process.

Factor	Type of Counterions	Dissociation Temperature (°C)	Concentration of Counterions (mol/L)	Ratio of Acetonitrile (%)	Concentration of MPH (mg/mL)
Type of Counterions	H^+^	30	1	0	0.1
	K^+^	30	1	0	0.1
	Na^+^	30	1	0	0.1
Concentration of Counterions	K^+^	30	1	0	0.1
	K^+^	30	1.5	0	0.1
	K^+^	30	2.5	0	0.1
Ratio of Acetonitrile	K^+^	30	2.5	0	0.1
	K^+^	30	2.5	20	0.1
Dissociation Temperature	K^+^	30	2.5	20	0.1
	K^+^	40	2.5	20	0.1
	K^+^	50	2.5	20	0.1

**Table 3 polymers-15-01191-t003:** Items of analytical method validation.

Items of Validation	Sample Preparation	Standards
Specificity	Blank solvent, blank excipients, reference solution, test solution, degradation products of acid (add 1.2 M HCl, 25 °C for 4 h), degradation products of base (add 1.2 M NaOH, 25 °C for 10 min), degradation products of oxygen (add 30% H_2_O_2_, 25 °C for 4 h)	Peak separation > 1.5, peak purity > 980
Linearity and range	MPH reference solution of 0.06 mg/mL–0.32 mg/ml	r > 0.999
Accuracy (n = 3)	MPH reference solutions of 0.16 mg/mL, 0.2 mg/mL, and 0.24 mg/mL were added to blank excipients	Recoveries in the range of 95.0%–105.0%, RSD ≤ 2.0%
Repeatability (n = 6)	Six reference solutions and test solutions of MPH	RSD ≤ 2.0%
Intermediate precision (n = 12)	Six reference solutions and test solution of MPH (2 analysts)	RSD ≤ 2.0%

**Table 4 polymers-15-01191-t004:** Statistical analysis of extraction methods.

Extraction Method	*p*-Values
Ultrasonic extraction	0.0016 (<0.05)
High-pressure homogenization	0.0017 (<0.05)
Ultrasonic cell crusher	0.0015 (<0.05)
Dissociation	-----

**Table 5 polymers-15-01191-t005:** Statistical analysis of factors influencing the dissociation process.

Factors Influencing the Dissociation Process		*p*-Values
Concentration of counterions	1 mol/L K^+^	0.0275 (<0.05)
	1.5 mol/L K^+^	0.0306 (<0.05)
	2.5 mol/L K^+^	-
Ratio of acetonitrile	0%	0.0030 (<0.05)
	20%	-
Dissociation temperature	30 °C	0.0289 (<0.05)
	40 °C	0.8509 (>0.05)
	50 °C	-

**Table 6 polymers-15-01191-t006:** Results of analytical method validation.

Items of Validation	Results	Standards
Linearity and range	Y = 0.6461X + 0.7529 (r = 0.9999)	r > 0.999
Accuracy	Recoveries = 100.6% RSD = 1.56%	Recoveries in the range of 95.0–105.0%, RSD ≤ 2.0%
Repeatability	RSD = 0.17%	RSD ≤ 2.0%
Intermediate precision	RSD = 0.39%	RSD ≤ 2.0%

**Table 7 polymers-15-01191-t007:** Thermodynamics parameters at different temperatures.

T(K)	303	313	323
*Ke*	0.0280	0.0429	0.0504
Δ*G* (kJ/mol)	9.00	8.19	8.03
Δ*H* (kJ/mol)	2.7 × 10^−4^	2.7×10^−4^	2.7 × 10^−4^
Δ*S* (J/mol)	−29.72	−26.17	−24.85

**Table 8 polymers-15-01191-t008:** Fitting results of the dissociation reaction kinetic curve at 298 K.

Reaction Type	Kinetic Equation	Regression Equation	r
Zero-order	*Q_e_ − Q_t_ = −kt*	y = −1.9077x + 28.989	0.9347
First-order	*ln (Q_e_ − Q_t_) = −kt*	y = −0.1783x + 3.7309	0.9820
Second-order	*1/Q_e_ − Q_t_ = kt*	y = 0.0226x − 0.0417	0.9963

**Table 9 polymers-15-01191-t009:** The linear regression analysis for HDM, SCM, and Boyd.

Kinetic Model	Rate-Limiting Step	Y-Axis, F(x)	X-Axis	r
SCM	a. Film diffusion	*F*	t	0.9347
	b. Matrix diffusion	$3-3{\left(1-F\right)}^{2/3}-2F$	t	0.9695
	c. Chemical reaction	$1-{\left(1-F\right)}^{1/3}$	t	0.9731
HDM	a. Film diffusion	−ln(1−F)	t	0.9868
	b. Matrix diffusion	$-\ln\left(1-F^{2}\right)$	t	0.9881
Boyd	a. Matrix diffusion	$-2.303\log_{10}\left(1-F\right)-0.498$	t	0.9870

**Table 10 polymers-15-01191-t010:** Boyd model fitting on type of counterions.

Factors	Value	Regression Equation	r
Type of Counterions	1 mol/L H^+^	y = 0.0711x + 1.4717	0.9864
	1 mol/L Na^+^	y = 0.0401x + 0.7082	0.9982
	1 mol/L K^+^	y = 0.0792x + 0.8858	0.9831

**Table 11 polymers-15-01191-t011:** Boyd model fitting on the concentration of counterions.

Factors	Value	Regression Equation	r
Quantity of Counterions	1 mol/L K^+^	y = 0.1153x + 0.5547	0.9999
	1.5 mol/L K^+^	y = 0.0838x + 0.7327	0.9970
	2.5 mol/L K^+^	y = 0.0469x + 0.4513	0.9977

**Table 12 polymers-15-01191-t012:** Boyd model fitting on the ratio of acetonitrile.

Factors	Value	Regression Equation	r
Ratio of acetonitrile	0%	y = 0.0469x + 0.4513	0.9977
	20%	y = 0.1817x + 0.4575	0.9974

**Table 13 polymers-15-01191-t013:** Boyd model fitting on dissociation temperature.

Factors	Value	Regression Equation	r
Dissociation temperature	30 °C	y = 0.1944x + 0.3198	0.9998
	40 °C	y = 0.1817x + 0.4575	0.9974
	50 °C	y = 0.1931x + 1.0388	0.9920

## Data Availability

No new data were created or analyzed in this study. Data sharing is not applicable to this article.

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
