# Peer review of "Methods and Characteristics of Drug Extraction from Ion-Exchange-Resin-Mediated Preparations: Influences, Thermodynamics, and Kinetics"

_polymers, 2023, doi:10.3390/polym15051191_

Round 1

Reviewer 1 Report

I agree fully with publishing of this paper no finding some mistakes.

Author Response

Response to Reviewer 1 Comments

Point 1: I agree fully with publishing of this paper no finding some mistakes.

Response 1: We appreciate the reviewer’s positive comments on our work.

Reviewer 2 Report

The article is devoted to the method of drug extraction of ion-exchange resin mediated preparations. It is shown that this method has a high efficiency. Authors should clearly describe the purpose and novelty of the study in the introduction.

Author Response

Response to Reviewer 2 Comments

Point 1: The article is devoted to the method of drug extraction of ion-exchange resin mediated preparations. It is shown that this method has a high efficiency. Authors should clearly describe the purpose and novelty of the study in the introduction.

Response 1: Thank you very much for your suggestions. We have made correction according to the reviewer's comments in lines 176-185, 352-360.

Reviewer 3 Report

The manuscript polymers-2071871 provides a comparison between different methods for drug extraction from ion exchange resin preparations and, after selection of the best extraction method, analyses how the type and concentration of counter ion, the addition of solvents and temperature influence the extraction. Finally, a thermodynamic and kinetic characterization of the model is presented.

Specific Comments

The introduction doesn’t provide the sufficient scientific background.

The level of English writing is poor and the whole document should be read by a native speaker or, at least, by someone with more knowledge regarding the English language.

There is no reference to the way how data analysis was performed.

Graphics are extremely basic, without any statistical information beyond the mean values. There is no reference to the number of experiments that were performed, no indication of standard deviation values, confidence intervals or any statistical parameter that may help the reader to understand the significance of the results.

Figure 6 doesn’t have a legend.

Regarding the analytical method validation, despite the reference to the RSD, there is no indication of the number of experiments that were performed. The precision tests that were performed are repeatability tests. No experiments regarding intermediate precision were presented. Also, no reference was made regarding the methodology used in the degradation tests that were performed.

The discussion section is very poor, and the results are not sufficiently cross-checked and analysed in comparison with the existing literature. This section has 1,5 pages length and there is not a single reference to the bibliography for a whole page.

Finally, I think that this document needs a complete overhaul and in the present form doesn’t present the quality needed to be published in an international and peer reviewed journal.

Round 2

Reviewer 3 Report

The document suffered a major revision ans has now a minimal quality for publication. However, graphics are should still be improved. For instance, some bar graphs could be grouped in just one figure and not use one figure for each one. Also, the numbers on the graphs are many times crossed by the deviation bars and that makes it difficult to read the numbers clearly. The authors refer that the mean values were used for analysis, but the mean values by themselves don'y have much value. The deviation must be considered as well. 

Author Response

Response to Reviewer 3 Comments

Point 1: The document suffered a major revision ans has now a minimal quality for publication. However, graphics are should still be improved. For instance, some bar graphs could be grouped in just one figure and not use one figure for each one. Also, the numbers on the graphs are many times crossed by the deviation bars and that makes it difficult to read the numbers clearly. The authors refer that the mean values were used for analysis, but the mean values by themselves don'y have much value. The deviation must be considered as well.

Response 1: Thank you for your constructive suggestions. We have made correction according to the reviewer's comments in lines 121,193-293. We used t-tests for data analysis.